# Pharmacologic neuroprotective agents for the treatment of perinatal asphyxia in low-income and lower-middle-income countries: A systematic review and meta-analysis of randomised controlled trials

Victor Ayodeji Ayeni[1]*, Aisha Adam Abdullahi[2], Abdulmuminu Isah[3],
Love Bukola Ayamolowo[4], Timothy Adeyemi[5], Motunrayo Adebukunola Oladimeji[6],
Ufuoma Shalom Ahwinahwi[7], Vitalis Ikenna Chukwuike[8], Oluchukwu Perpetual Okeke[9],
Olunike Rebecca Abodunrin[10], Folahanmi Tomiwa Akinsolu[11,9],
Olajide Odunayo Sobande[12]

1 Department of Paediatrics, Babcock University, Ilishan-Remo, Nigeria, 2 Department of Epidemiology and Population Health, Kano Independent Research Centre Trust (KIRCT), Kano State, Nigeria, 3 Department of Clinical Pharmacy and Pharmacy Management, University of Nigeria, 4 Department of Nursing Science, Obafemi Awolowo University, Ile-Ife, Nigeria, 5 Department of Physiotherapy, Bowen University, Iwo, Osun State, Nigeria, 6 Department of Anaesthesia, Lagos University Teaching Hospital, Idi-Araba, Lagos, 7 Department of Pharmacy, Delta State University, Abraka, Nigeria, 8 International Institute of Biomedical Tissue Engineering and Regenerative Medicine Research Unit, Department of Industrial and Medicinal Chemistry, David Umahi Federal University of Health Sciences, Uburu, Ebonyi State, Nigeria, 9 Nigerian Institute of Medical Research Foundation, Yaba, Lagos State, Nigeria, 10 Department of Epidemiology and Biostatistics, Nanjing Medical University, Nanjing, China, 11 Lead City University, Department of Public Health, Ibadan, Nigeria, 12 Nigerian Institute of Medical Research Foundation, Yaba, Lagos State, Nigeria

* tioluwa@hotmail.com

## Abstract

### Background

Perinatal asphyxia (PA) is a major contributor to neonatal mortality and long-term neurodevelopmental impairment, particularly in low- and middle-income countries (LMICs), where the effectiveness of therapeutic hypothermia remains limited. Pharmacologic neuroprotective agents have shown potential as alternative treatments, but their efficacy in low-income and lower-middle-income countries (LILMICs) is not well established. This systematic review aimed to assess the effectiveness of pharmacologic interventions in neonates with PA in LILMICs.

### Methods

A systematic search of PubMed, Web of Science, CINAHL, and Google Scholar was conducted for randomised controlled trials (RCTs) published between 2000 and 2024. Eligible studies compared pharmacologic neuroprotective agents with placebo or standard care, excluding therapeutic hypothermia, among neonates diagnosed with

**Data availability statement:** All relevant data are within the paper and its Supporting Information files.

**Funding:** VAA, AAA, AI, LBA, TA, MAO, USA, VIC got a grant from Nigerian Institute of Medical Research Foundation for the work. Grant Number: NF-GMTP-24-152809 The funders had no role in study design, data collection and analysis, decision to publish, or preparation of the manuscript.

**Competing interests:** The authors have declared that no competing interests exist.

PA in LILMICs. Data on survival and neurodevelopmental outcomes were extracted and synthesized; meta-analyses were conducted where appropriate.

## Results

Twelve RCTs involving 1,008 neonates were included. The majority (91.7%) of studies were conducted in Asia, with only one study from Africa. Magnesium sulphate was the most frequently evaluated agent (66.7% of studies), followed by melatonin, topiramate, erythropoietin, and citicoline. Melatonin was associated with improved survival, and all agents showed improved short-term neurological outcomes. Neurodevelopmental outcomes at 3, 6, 12, and 19 months were generally favourable, though data remained limited.

## Conclusion

Pharmacologic neuroprotective agents show promise in improving survival and neurological outcomes in neonates with PA in LILMICs. However, more robust, multi-center RCTs are needed to confirm their efficacy and establish them as feasible alternatives to therapeutic hypothermia in these settings.

## Background

Neonatal mortality remains a major contributor to under-five mortality globally, with the burden disproportionately higher in low and middle-income countries (LMICs) [1]. Sub-Saharan Africa (SSA) accounts for 38% of global neonatal deaths, largely from preventable causes such as perinatal asphyxia (PA) [2]. In high-income countries, the prevalence of PA is approximately 2 per 1,000 live births, whereas in LMICs, rates are up to ten times higher due to limited access to skilled care during childbirth [3]. Among neonates affected by PA, 15–25% die in the neonatal period, and about 25% of survivors suffer long-term neurological impairments [4]. SSA alone accounts for nearly half (46%) of global PA-related deaths and a significant proportion of associated neurodevelopmental disabilities [2].

PA is defined as impaired gas exchange or compromised blood flow around the time of birth, leading to persistent hypoxemia and hypercarbia [5]. This often progresses to neonatal encephalopathy, which imposes severe short- and long-term consequences on the child, family, and healthcare systems, particularly in resource-limited settings [2]. In these contexts, the economic burden further compounds the challenges of managing PA effectively.

While Sustainable Development Goal 3.2 prioritises the reduction of preventable neonatal deaths [6], effective strategies for the prevention and treatment of PA remain limited in LMICs. Prevention is key in the management of PA. However, the training of health workers on resuscitation skills has only been partially effective in low-income countries and lower-middle-income countries [7], with resultant high rates of morbidity and mortality.

Therapeutic hypothermia (TH) is the current standard of care in high-income countries and has demonstrated neuroprotective efficacy in multiple randomised controlled trials [8,9]. However, its implementation in low-income and lower-middle-income countries (LILMIC) is limited due to infrastructure and cost [10]. Moreover, emerging evidence suggests that TH may be ineffective—or even harmful—in these settings. The HELIX trial, conducted across tertiary neonatal units in India, Sri Lanka, and Bangladesh, found no benefit of TH in LILMICs [11], a finding supported by meta-analyses of similar trials [12]. Contributing factors may include a higher prevalence of low birthweight and small-for-gestational-age neonates, as well as earlier and more severe hypoxic insults [10,11].

Supportive care (e.g., fluid therapy, seizure control, nutritional support) remains the mainstay of treatment in most LMICs but has not significantly improved long-term neurodevelopmental outcomes [13–15]. This highlights the urgent need to explore alternative, cost-effective neuroprotective therapies suitable for resource-constrained environments.

PA initiates a cascade of biochemical and physiological disturbances, including oxidative stress and anaerobic metabolism, leading to the generation of free radicals that cause cellular injury [16]. In response, several novel pharmacological agents with neuroprotective properties are being investigated—either as adjuncts to TH or as standalone therapies. These interventions target neuronal apoptosis, oxidative stress, and restoration of neurophysiology [16–18].

This study is aimed at synthesising the current evidence on neuroprotective alternatives to therapeutic hypothermia for the treatment of perinatal asphyxia and neonatal encephalopathy in human neonates in LILMICs.

## Study objectives

The primary objective that guided this systematic review was to assess the effectiveness of pharmacologic neuroprotective alternatives to therapeutic hypothermia for treating perinatal asphyxia in LILMIC.

The review addressed the following research questions:

1. What were the key pharmacologic agents that showed potential neuroprotective benefits for neonates with perinatal asphyxia in LILMIC?

2. What were the neurodevelopmental and survival outcomes associated with pharmacologic neuroprotective interventions in neonates with perinatal asphyxia in LILMIC?

3. How did pharmacologic alternatives compare to standard supportive care in reducing brain injury and preventing neurodevelopmental sequelae among neonates affected by perinatal asphyxia/neonatal encephalopathy?

## Methods

### Protocol registration

This systematic review was registered on PROSPERO with the ID: CRD42024589526. The review sought to provide a broad overview of the available literature on neuroprotective interventions and identify research gaps in this area. It adhered to the Preferred Reporting Items for Systematic Reviews and Meta-Analyses (PRISMA) guidelines, ensuring methodological rigour and transparency [19]. (Supplementary File 1)

Using the PICO framework recommended for an exhaustive literature search [20,21], Table 1 provides a structured approach to comparing neonates with perinatal asphyxia, pharmacologic neuroprotective interventions, and key outcomes, with a placebo group serving as a control for the study's focus on mortality and neurodevelopmental results.

**Table 1. PICO framework.**

| P – population | I – Intervention | C – comparator | O – Outcome |
|---|---|---|---|
| Neonates with perinatal asphyxia | Pharmacologic neuroprotective agents | Supportive care with or without a placebo | Survival, early neurologic outcomes and late neurodevelopmental outcomes |

## Search strategy

The search was conducted to identify relevant studies on pharmacologic neuroprotective alternatives to therapeutic hypothermia. The databases searched included PubMed, CINAHL, Web of Science, and Google Scholar. A comprehensive search strategy was developed, which included the following keywords and their synonyms: "Perinatal asphyxia" OR "Neonatal encephalopathy" AND "neuroprotect*" OR "Magnesium sulphate" OR "Erythropoietin" OR "Melatonin" OR Topiramate, etc. The search strategy is in Supplementary File 2. Additional searches were performed on ClinicalTrials.gov to include registered trials. The review covered studies published between January 2000 and August 2024. The PRISMA flow chart of the search process is shown in Fig 1.

## Selection criteria

The inclusion criteria covered studies that:

1. Investigated pharmacologic neuroprotective interventions started within 24 hours of birth.

2. Were conducted in low-income or lower-middle-income settings,

3. reported neurodevelopmental outcomes, survival rates, or time to recovery from neurological depression or adverse events.

4. Compare pharmacologic alternatives to standard care and/or placebo (other than therapeutic hypothermia).

5. Were randomised controlled trials

## The exclusion criteria covered studies that

1. Compared Therapeutic hypothermia with other alternatives,

2. Were animal studies, in vitro studies, reviews without original data, or case reports.

## Study selection and screening

Search results from the databases were exported into Rayyan software, where duplicates were removed. Two independent reviewers with an acceptable Kappa score of 1.0 during a pretest (VAA and AAA) screened the titles and abstracts using the predefined inclusion and exclusion criteria. All conflicts were discussed by the reviewers and resolved appropriately. Full texts of potentially eligible studies were retrieved for further review.

## Data extraction

Data was extracted independently by VAA and AI using a standardised extraction form to ensure consistency. (Supplementary File 3) Study characteristics were collected from each included article, such as the country and region where the study was carried out, sample size, estimated gestational age (EGA), neuroprotective agent, dosage and route of administration, Control/placebo, and study design (Table 2). The different outcomes measured, including survival, the duration from birth to resolution of immediate neurological depression, and short- and long-term neurodevelopmental outcomes (survival; successful initiation of oral feeding at discharge; mean time to initiation of enteral feeds; need for seizure control at review time; mean/median number of days to seizure control; duration of hospital stay; neuroimaging features at discharge/ 1 month of age; neurologic outcome at discharge/ 1 month of age; neurodevelopmental outcome at 3 months, 6 months, 9 months and 11 months; adverse event), were also extracted (Tables 3–4).

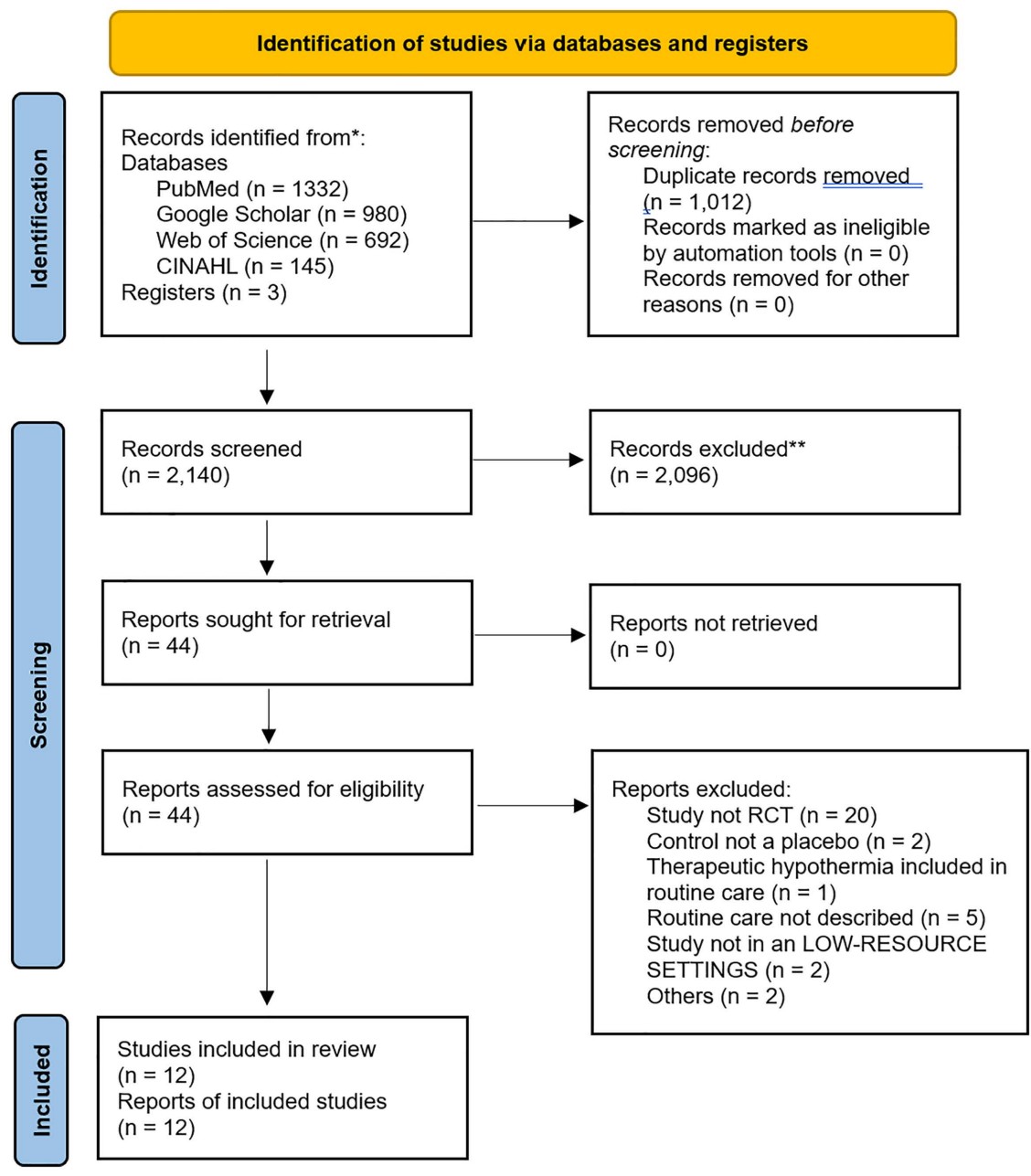

**Fig 1. PRISMA Flowchart for Pharmacologic Neuroprotective Alternatives to Therapeutic Hypothermia.**

## Quality assessment

The risk of bias for the randomised controlled trials (RCTs) included in this review was assessed using the Joanna Briggs Institute (JBI) checklist [22]. (Supplementary File 4). Each article was evaluated on relevant domains, with a score of 1 assigned for "yes" and 0 for both "no" and "unclear." Domains deemed not applicable to a particular study were marked as "N/A." VAA, AAA, AI and LBA conducted the assessment collaboratively, with disagreements resolved through discussion until a consensus was reached. Studies with an aggregate score ranging from 0.7 to 1.0 were classified as having a low

Table 2. Characteristics of included studies.

| Ref. | Authors | Title | Country | Region | Sample Size | EGA* | Neuroprotective agents | Dosage | Route | Control | Quality |
|---|---|---|---|---|---|---|---|---|---|---|---|
| 1. | Ahmad et al., 2018 [31] | Role of melatonin in management of hypoxic-ischaemic encephalopathy in new-borns: A randomized control trial | Pakistan | Asia | 80 | Late pre-term and term | Melatonin | 10 mg* (single dose) | Enteral | Only supportive care | Low risk |
| 2. | Akther et al., 2020 [32] | Role of Topiramate in Moderate to Severe Perinatal Asphyxia – A Randomized Controlled Clinical Trial | Bangladesh | Asia | 64 | Term | Topiramate | 10mg daily for 3 days | Enteral | Only supportive care | Low risk |
| 3. | Bhat et al., 2009 [33] | Magnesium sulphate in severe Perinatal Asphyxia: A Randomized, Placebo-Controlled Trial | India | Asia | 40 | Term | Magnesium Sulphate | 250mg/kg per dose daily | IV infusion | Normal saline | Low risk |
| 4 | Iqbal et al., 2021 [34] | The Neuroprotective Efficacy of Postnatal Magnesium Sulfate in Term or Near-Term Infants With Moderate-to-Severe Birth Asphyxia | Pakistan | Asia | 62 | Term | Magnesium Sulphate | 250mg/kg per dose daily | IV infusion | Dextrose water | Low risk |
| 5. | Mahmood et al., 2015 [35] | Effect of Postnatal Magnesium Sulfate Infusion on Neurological Outcome of Term Neonates with Severe Perinatal Asphyxia | Pakistan | Asia | 64 | Term | Magnesium Sulphate | 250mg/kg/ dose daily | IV infusion | Normal saline | Moderate risk |
| 6. | Malla et al., 2017 [36] | Erythropoietin monotherapy in perinatal asphyxia with moderate to severe encephalopathy: a randomized placebo-controlled trial | India | Asia | 100 | Term | Erythropoietin | 500 IU/kg/ dose on alternate days | IV | Normal saline | Low risk |
| 7. | Prakash et al. 2016 [37] | Neurodevelopmental Outcome at 12 Months of Postnatal Magnesium Sulphate Therapy for Perinatal Asphyxia | India | Asia | 120 | Term | Magnesium Sulphate | 250mg/kg/ dose | IV infusion | Normal saline | Low risk |
| 8. | Rahman et al. 2021 [38] | Effectiveness of Magnesium Sulphate In Term Neonate With Perinatal Asphyxia: A Study In Faridpur Medical College Hospital, Faridpur, Bangladesh | Bangladesh | Asia | 50 | Term | Magnesium Sulphate | 250mg/kg/ dose | IV infusion | Normal saline | Low risk |
| 9. | Rashid et al.,2015 [39] | Role of Magnesium Sulphate In Short Term Neurological Outcome of Perinatal Asphyxia | Pakistan | Asia | 200 | Term | Magnesium Sulphate | 250mg/kg/ dose | IV infusion | Normal saline | Moderate risk |
| 10. | Sajid et al., 2018 [17] | Therapeutic Efficacy of Magnesium Sulphate On Neurological Outcome of Neonates With Severe Birth Asphyxia | Pakistan | Asia | 66 | Term | Magnesium Sulphate | 250mg/kg/ dose | IV infusion | Normal saline | Low risk |
| 11. | Salamah et al., 2023 [40] | Citicoline in hypoxic-ischemic encephalopathy in neonates: a randomized controlled trial | Egypt | Africa | 80 | Late pre-term and Term | Citicoline | 10mg/kg | IV | Placebo | Low risk |
| 12. | Siddiqui and Butt, 2021 [41] | Role of Intravenous Magnesium Sulphate in Term Neonates with Hypoxic Ischemic Encephalopathy (HIE) in a Low-income Country: A Randomised Clinical Trial | Pakistan | Asia | 82 | Term | Magnesium Sulphate | 250mg/kg/ dose | IV infusion | Normal saline | Low risk |

* EGA: Estimated Gestational Age; IV: Intravenous; mg: milligram; kg: Kilogram.

**Table 3. Descriptive Characteristics of Outcomes.**

| Authors/ Year | Intervention | Outcome 1 IG (%) | Outcome 1 CG (%) | Outcome 2 IG (%) | Outcome 2 CG (%) | Outcome 3 IG (days) | Outcome 3 CG (days) | Outcome 4 IG (%) | Outcome 4 CG (%) | Outcome 5 IG (IQR) (days) | Outcome 5 CG (IQR) (days) | Outcome 6 IG (SD) (days) | Outcome 6 CG (SD) (days) | Outcome 7 IG (%) | Outcome 7 CG (%) | Outcome 8 IG (%) | Outcome 8 CG (%) | Outcome 9 |
|---|---|---|---|---|---|---|---|---|---|---|---|---|---|---|---|---|---|---|
| Ahmad et al., 2018 | Melatonin | 87.5 | 65 | – | – | – | – | – | – | – | – | – | – | – | – | – | – | Not reported |
| Akther et al., 2020 | Topiramate | – | – | – | – | 2.63±1.1 | 4.5±1.3 | – | – | 24 (19.536) | 72 (24,72) | 6.3±1.7 | 10.3±2.7 | 93 | 52 | 73 | 41 | Not reported |
| Bhat et al., 2009 | MgSO$_4$ | 90 | 90 | 77 | 37 | – | – | 83 | 56 | – | – | – | – | 83 | 56 | 77 | 44 | Apnoea |
| Iqbal et al., 2021 | MgSO$_4$ | 93.5 | 87.1 | | | 1.58±0.56 | 2.52±0.96 | – | – | 1.71±0.46 | 2.65±1.11 | 3.26±1.06 | 4.39±1.99 | 57.1 | 39.3 | – | – | Not reported |
| Mahmood et al., 2015 | MgSO$_4$ | – | – | 71.4 | 40 | – | – | – | – | – | – | – | – | 88.6 | 62.9 | – | – | Not reported |
| Malla et al., 2017 | Erythropoietin | 16 | 16 | – | – | 6±4.3 | 9.7±5.1 | 81 | 57 | – | – | 9.7±6.9 | 13.5±8.1 | 60 | 40 | – | – | Transient increase in red cell indices |
| Prakash et al., 2016 | MgSO$_4$ | 86.4 | 73.7 | – | | – | – | – | – | – | – | – | – | – | – | – | – | None |
| Rahman et al., 2021 | MgSO$_4$ | 80 | 76 | | | – | – | – | – | – | – | – | – | – | – | – | – | Not reported |
| Rashid et al., 2015 | MgSO$_4$ | – | – | 72 | 31 | – | – | – | – | – | – | – | – | – | – | – | – | Not reported |
| Sajid et al., 2018 | MgSO$_4$ | – | – | 75.7 | 39.4 | – | – | – | – | – | – | – | – | 84.9 | 51.5 | 75.8 | 45.4 | Not reported |
| Salamah et al., 2023 | Citicoline | – | – | – | – | – | – | 95 | 72.5 | – | – | 25.58±9.36 | 25.58±9.36 | 75 | 47.5 | – | – | Transient diarrhoea |
| Siddiqui & Butt, 2021 | MgSO$_4$ | 75 | 65 | 67.5 | 40 | – | – | 24.39 | 19.51 | – | – | – | – | – | – | 65 | 37.5 | None |

Outcome 1 – Survival; Outcome 2 – successful initiation of oral feeding at discharge; Outcome 3 – mean time to initiation of enteral feeds; Outcome 4 – no need for seizure control at review time; Outcome 5 – mean/median number of days to seizure control; Outcome 6 – duration of hospital stay; Outcome 7 – normal neuroimaging features at discharge/ 1 month of age; Outcome 8 – normal neurologic outcome at discharge/ 1 month of age; Outcome 9 – adverse event

IG – intervention group, CG – control group; SD standard deviation; IQR (interquartile range).

**Table 4. Inferential Characteristics of Outcomes.**

| Authors and Year of Publication | Intervention | Outcome 1 P val. | Outcome 1 RR (95% CI) | Outcome 2 P val. | Outcome 2 OR (95% CI) | Outcome 3 P val. | Outcome 4 P val. | Outcome 4 RR | Outcome 5 P val. | Outcome 6 P val. | Outcome 7 P val. | Outcome 7 OR/RR (95% CI) | Outcome 8 P val. | Outcome 8 OR (95% CI) |
|---|---|---|---|---|---|---|---|---|---|---|---|---|---|---|
| Ahmad et al., 2018 | Melatonin | 0.03* | | | | | | | | | | | | |
| Akther et al., 2020 | Topiramate | | | | | < 0.001 | | | 0.00 | 0.00 | 0.00 | OR 0.08 (0.02–0.39) | 0.02 | 0.14 (0.04–0.49) |
| Bhat et al., 2009 | MgSO$_4$ | 1 | | 0.02* | 0.18 [0.04–0.78] | | 0.1 | | | 0.07 | | | 0.02 | 5.5 (1.2–23.6) |
| Iqbal et al., 2021 | MgSO$_4$ | 039* | | | | 0.02 | | | 0.001 | 0.003 | 0.783 | | | |
| Mahmood et al., 2015 | MgSO$_4$ | | | 0.008* | | | | | | | | | | |
| Malla et al., 2017 | Erythropoietin | 0.61 | | 0.75 | | 0.006 | 0.002 | 0.46 (0.25–0.80) | | 0.04 | 0.012 | | | |
| Prakash et al. 2016 | MgSO$_4$ | 0.32 | 0.51 (0.14-1.88) | | | | | | | | 0.4 | RR 0.66 (0.42–1.03) | | |
| Rahman et al. 2021 | MgSO$_4$ | 1 | | | | | | | | | | | | |
| Rashid et al., 2015 | MgSO$_4$ | | | 0.001* | | | | | | | | | | |
| Sajid et al., 2018 | MgSO$_4$ | | | 0.002* | | | | | | | 0.003 | | 0.011 | |
| Salamah et al., 2023 | Citicoline | | | | | | 0.006* | | | 0.001 | 0.001 | | | |
| Siddiqui & Butt, 2021 | MgSO$_4$ | 0.329 | | 0.004* | | | 0.576 | | | | | | 0.007 | |

OR – Odd Ratio, RR – Relative Risks; Outcome 1 – Survival; Outcome 2 – successful initiation of oral feeding at discharge; Outcome 3 – mean time to initiation of enteral feeds; Outcome 4 – no need for seizure control at discharge; Outcome 5 – mean/median number of days to seizure control; Outcome 6 – duration of hospital stay; Outcome 7 – normal neuroimaging features at discharge/ 1 month of age; Outcome 8 – normal neurologic outcome at discharge/ 1 month of age. CI – confidence intervals.

risk of bias; those scoring between 0.5 and 0.69 were considered moderate, and any study scoring below 0.5 had a high risk of bias.

## Data analysis

The data synthesis of this review was focused on evaluating the effectiveness of pharmacologic neuroprotective interventions in improving neurodevelopmental outcomes and reducing mortality among neonates with PA in LILMIC. The extracted data were synthesised both quantitatively and qualitatively.

Meta-analysis was conducted using R Studio version 4.2.2 leveraging the 'meta' and 'metafor' packages, which provide robust tools for advanced meta-analytic procedures. Risk ratios (RR) and 95% confidence intervals (CIs) were calculated for dichotomous outcomes, such as mortality and neurodevelopmental impairments. RR was chosen as the effect measure due to its ease of interpretation in clinical contexts where outcomes are event-based.

To account for potential variations across studies—such as differences in study design, population characteristics, and intervention implementation, a random-effects model was adopted using the DerSimonian and Laird (DL) estimator [23]. This approach assumes that true effect sizes vary across studies, making it suitable for the heterogeneous settings often found in LILMICs. The DL method remains widely accepted in meta-analysis because of its simplicity and reliable performance across a broad range of applications.

Heterogeneity between studies was assessed using Cochran's Q test and the I² statistic. The I² statistic was used to quantify the proportion of variability due to heterogeneity rather than chance [24]. In line with Cochrane guidelines, an I² value greater than 50%, along with a Q test p-value below 0.01, was interpreted as evidence of substantial heterogeneity [24–26].

## Sensitivity analysis

To assess the robustness of the pooled effect estimates and evaluate the influence of potentially missing studies, a sensitivity analysis was conducted using the trim-and-fill method. This nonparametric approach estimates the impact of unpublished or missing studies on the overall effect size and adjusts the results accordingly to provide a more balanced interpretation [27].

## Subgroup analysis

Predefined subgroup analyses were carried out to explore sources of heterogeneity. These included comparisons based on the type of pharmacologic neuroprotective agent—such as erythropoietin, magnesium sulfate, and melatonin—allowing assessment of differences in treatment efficacy by drug class or mechanism of action. Additionally, studies were grouped according to neurologic follow-up periods, such as immediate neonatal outcomes versus outcomes measured at 6–12 months. This time-based stratification was necessary to capture delayed-onset neurodevelopmental sequelae that might not be evident in the early neonatal period.

## Publication bias

Publication bias was assessed through both visual and statistical methods. Funnel plots were examined for asymmetry, and Egger's regression test was applied to statistically detect small-study effects [28]. A significant Egger's test result (p<0.05), combined with funnel plot asymmetry, was considered indicative of potential publication bias. The use of both methods improves the sensitivity and reliability of bias detection [29].

## Assessment of the certainty of a body of evidence

The certainty of the body of evidence was assessed using the GRADE (Grading of Recommendations Assessment, Development and Evaluation) framework [30] across the domains of risk of bias, inconsistency, indirectness, imprecision and publication bias.

## Results

### Selection of studies

Fig 1 presents the flowchart of the data search, which resulted in the retrieval of 3,152 records. After removing duplicates, 2,140 remained for eligibility screening based on title and abstract. Of these, 2,096 studies were excluded. After reviewing the full-text record of 44 studies, 12 met the inclusion criteria and were included in the review [17,31–41]. The 32 excluded at the final stage are reported in the Supplementary file 5 with the reason for their exclusion.

Table 2 reflects the characteristics of the studies included in the review from 2009 [33] to 2024 [40]. Randomised controlled trials that investigated the effectiveness of pharmacologic neuroprotective alternatives to therapeutic hypothermia for treating perinatal asphyxia were few in LILMIC between 2005 and 2014, with only one (8.3%) study [33]. However, from 2015 to 2024, there was a substantial increase in research output, with 11 studies (91.7%) published during this period [17,31,32,34–41]. Sample sizes across the studies varied widely from 40 [33] to 200 [39] participants, with an aggregate of 1008.

### Geographical distribution of the studies

The research had a regional imbalance, with most of the included studies focused on Asia, 11 (91.7%) [17,31–39,41]. In Asia, Pakistan contributed the most studies, 6 (50%) [17,31,34,35,39,41], with the rest being three (25%) in India [33,36,37] and two (16.7%) in Bangladesh [32,38]. Only one study (8.3%) was conducted in Africa, specifically Egypt [40].

### Characteristics *of* neuroprotective agents and control

The most frequently studied neuroprotective agent was magnesium sulphate (66. 6%) with a standard dosage of 250 mg/kg per dose [17,33–35,37–39,41]. Other agents explored in the studies included melatonin administered at 10 mg (as a single dose) [31], topiramate at 10 mg daily [32], erythropoietin at 500 IU/kg on alternate days [36], and citicoline at 10 mg/kg [40], each of which was investigated in one study (8.3%). Regarding the route of administration, the majority of the studies (83%) used neuroprotective agents with an intravenous route, which includes magnesium sulphate, citicoline and erythropoietin [17,33–41]. Two studies (16.7%) focused on melatonin [31] and topiramate [32] used oral administration as the route. The majority of the studies (66.7%) used normal saline as a placebo [17,33,35–39,41], while Iqbal *et al.,* [34] used dextrose water as the control. In the selected studies, the intervention and control groups received comparable supportive routine care (which did not include therapeutic hypothermia) in addition to the intervention or placebo (Table 2).

### Outcomes

Table 3 shows the outcomes measured by the selected studies. Eight outcomes were common to at least two or more studies. These included mortality outcomes reported by 6 studies [31,33,34,36–38] and time interval from birth to recovery from initial neonatal encephalopathy (measured by the initiation of oral feeds, seizure control or duration of hospital stay), which was reported by nine studies [17,32–36,39–41]. Seven studies [18–22,26,27] reported neuroimaging features of residual brain damage at discharge or one month. In comparison, four studies [18,19,26,28] reported the neurological outcome at one month. The study by Prakash *et al.*[37] reported some of the outcomes of interest, but the authors grouped the outcomes under two subgroups (severe HIE and moderate HIE), so the outcomes could not be listed with the other studies (Table 3). Neurologic outcomes at 3 months, 6 months, 12 months and 19 months were reported by Akther *et al.* [32], Iqbal *et al.* [34], Prakash *et al.* [37], and Malla *et al.* [36], respectively.

The reviewed studies reported better neurological outcomes in neonates with perinatal asphyxia who received pharmacologic interventions compared to those who received standard supportive care. Table 4 shows the effects of the interventions on the different outcomes investigated by the included studies [23]. Only the study which employed melatonin [22] reported better survival rates for the survival outcome. On the earlier tolerability of feeds and successful initiation of feeds

by discharge time, magnesium sulphate [17,33,34,39,41], topiramate [32] and erythropoietin [36] were found to improve outcomes. Topiramate [32], erythropoietin [36] and citicoline [40] were the neuroprotective agents with better seizure control. All four neuroprotective agents reduced the period of hospital stay [32,34,36,40]. Topiramate [32] and citicoline [40] resulted in fewer abnormal neuroimaging findings, while magnesium sulphate [17,33,41] and topiramate [32] improved neurological outcomes at discharge or one month of age.

In addition, better neurological outcomes at 3 months were reported for the use of topiramate 24/28 (85.7%) versus 10/26 (38%) (OR 0.19, 95% CI 0.59–0.60) [32]. The neurologic outcome at 6 months [12/21 (77%) versus 8/19 (70.4%); p = 0.535] [34] and 12 months [19/22 (86.4%) versus 14/19 (73.7%); RR 0.51, 95% CI (0.14–1.88)] [37] were also better with the use of magnesium sulphate. Malla *et al.* [36] reported the longest follow-up among all the studies and reported a better neurodevelopmental outcome at 19 months with the use of erythropoietin [30/42 (71%) versus 15/42 (36%); RR 0.44 95% CI (0.25–0.76)]. (See Table 4)

Rahman *et al.* [38] subdivided the babies into moderate and severe HIE. For the subcategory of babies with moderate HIE, they reported earlier control of seizures (p = 0.04), earlier attainment of full oral feeds (p = 0.04), and shorter duration of hospital stay (p = 0.03) in the intervention group compared with the controls. However, this significant effect was not demonstrated in babies with severe HIE (p > 0.05).

**Melatonin** significantly improved survival (p = 0.03) [31]. **Topiramate** demonstrated significant benefits for seizure control (p < 0.001) and shorter hospital stays (p = 0.00), with improved neurological outcomes at one month (OR = 0.08, 95% CI: 0.02–0.39, p = 0.02) and 3 months [OR 0.19, 95% CI 0.59–0.60] [32]. **Magnesium sulphate** consistently showed earlier tolerability of enteral feeds (p = 0.02) [34], more successful initiation of sucking by discharge time across multiple outcomes, earlier feeding initiation (p < 0.05) [17,33,35,39], shorter mean time to seizure control (p = 0.001) [34], shorter hospital stays (p = 0.003) [26] and better neurological outcomes at one month (p < 0.05) [17,33,41], 6 months (p = 0.535] [34] and 12 months [RR 0.51, 95% CI (0.14–1.88)]. **Erythropoietin** demonstrated earlier tolerability of enteral feeds (p = 0.006), better seizure control (OR = 0.46, 95% CI: 0.25–0.80) and shorter duration of hospital stays (p = 0.04) and better neurodevelopmental outcomes at 19 months RR 0.44 95% CI (0.25–0.76) [36] **Citicoline** led to significantly better seizure control (p = 0.006) and shorter hospital stays (p = 0.001) than controls [40].

The included articles report no significant adverse effects with the pharmacologic therapeutic agents, apart from one out of the eight studies which studied magnesium sulphate that reported apnoea requiring ventilation in two infants (See Table 3). The overall methodological quality of the studies was deemed low risk of bias in 10 (83.3%), while only two studies were classified as moderate risk of bias.

## Meta-analysis

A meta-analysis was conducted to evaluate the effectiveness of pharmacologic neuroprotective alternatives in treating perinatal asphyxia, focusing on five key outcomes which were reported in frequencies and proportions: survival, successful initiation of oral feeding at discharge, absence of anticonvulsant requirements at discharge, normal neuroimaging features at one month, and normal neurologic/neurodevelopmental outcomes.

## Survival

A meta-analysis was performed on seven articles that reported survival rates in neonates with perinatal asphyxia. Our findings indicate that pharmacologic neuroprotective agents contribute to increased survival rates in neonates compared to the control group, with a RR:1.11 (95% CI: 1.01–1.22) and no significant heterogeneity (I² = 0%) (see Fig 2).

Additionally, survival was better with the use of melatonin, reported by Ahmad *et al.*, [33] (RR: 1.32; 95% CI; 1.03–1.68, I² = 0%) than magnesium sulphate (pooled RR: 1.08; 95% CI; 0.97–1.20, I² = 0%), when compared with the control groups (subgroup *p*-value = 0.14) (see Fig 3).

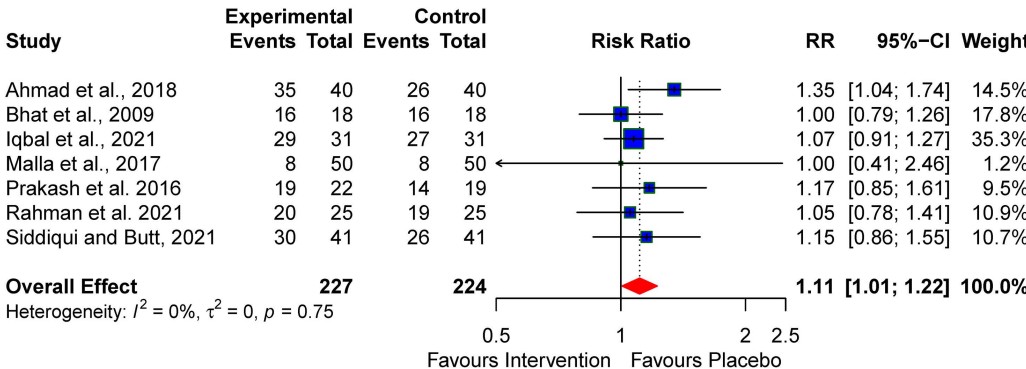

**Fig 2. Pooled Effect of the Neuroprotective Agents on Survival.**

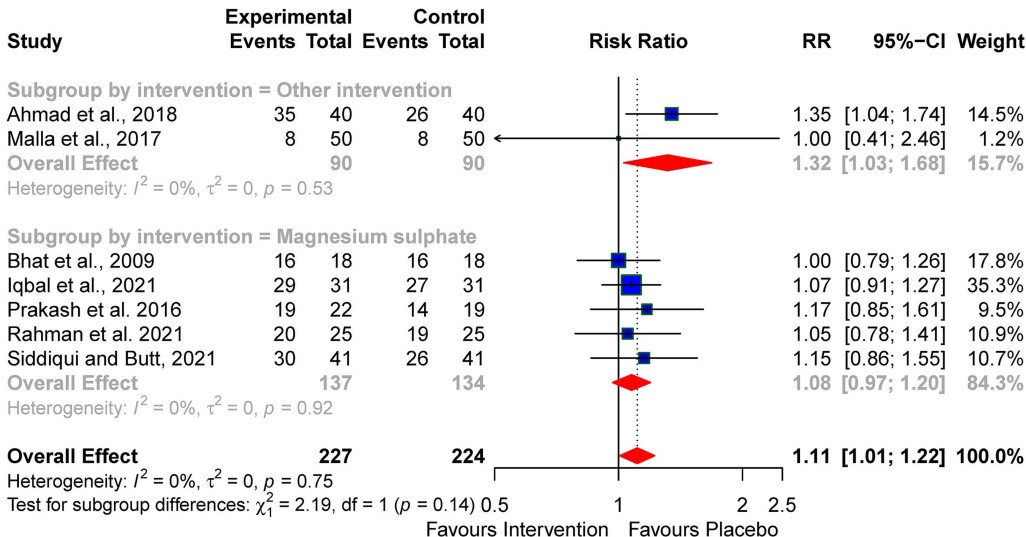

**Fig 3. Subgroup analysis of the effect of the neuroprotective agents on Survival.**

## Successful initiation of oral feeding at discharge

Five studies examined the successful initiation of oral feeding at discharge following pharmacologic interventions for treating perinatal asphyxia. Pooled analysis revealed a significant improvement in the initiation of oral feeding among neonates with perinatal asphyxia who received pharmacologic agents, with an RR of 1.66 (95% CI: 1.24–2.22). However, notable heterogeneity was observed ($I^2 = 84\%$) (see Fig 4).

On subgroup analysis, magnesium sulphate demonstrated better improvement in the successful initiation of oral feeding by discharge (pooled RR: 1.98; 95% CI; 1.63–2.40, $I^2 = 0\%$) compared to erythropoietin reported by Malla *et al*, [36] (RR: 1.02, 95 CI; 0.89–1.17) (subgroup *p*-value=<0.01) (see Fig 5).

## Absence of anticonvulsant requirement at discharge

Our findings indicate that the pharmacologic neuroprotective alternatives contribute to the absence of anticonvulsant requirements at discharge RR:1.34 (95% CI: 1.25–1.57) and no significant heterogeneity ($I^2 = 0\%$) (see Fig 6).

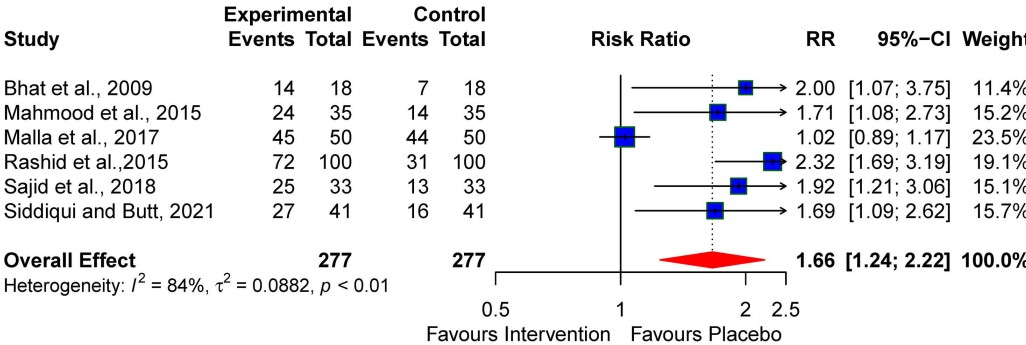

**Fig 4. Pooled effect of neuroprotective agents on successful initiation of oral feeds at discharge.**

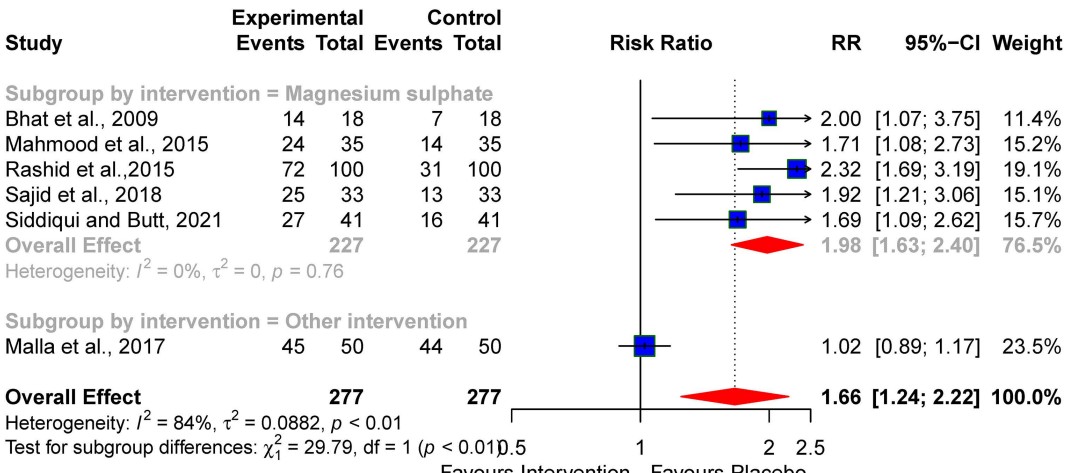

**Fig 5. Subgroup analysis of the effect of neuroprotective agents on successful initiation of oral feeds at discharge.**

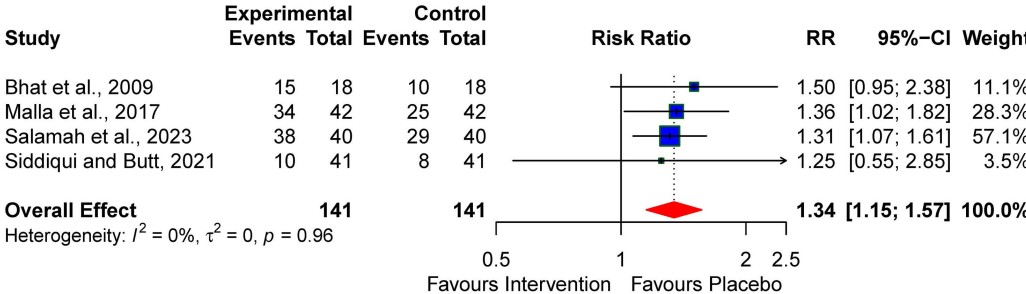

**Fig 6. Pooled effect of the pharmacological agent on the need for the anticonvulsant requirement at discharge.**

Subgroup analysis of magnesium sulphate showed a less need for anticonvulsants at discharge RR: 1.44(95% CI; 0.96–2.15, I²=0%) compared to the pooled RR for the other pharmacological agents (erythropoietin and citicoline), used in the studies that reported this outcome [33,36,40,41] (subgroup p-value=0.72) (see Fig 7).

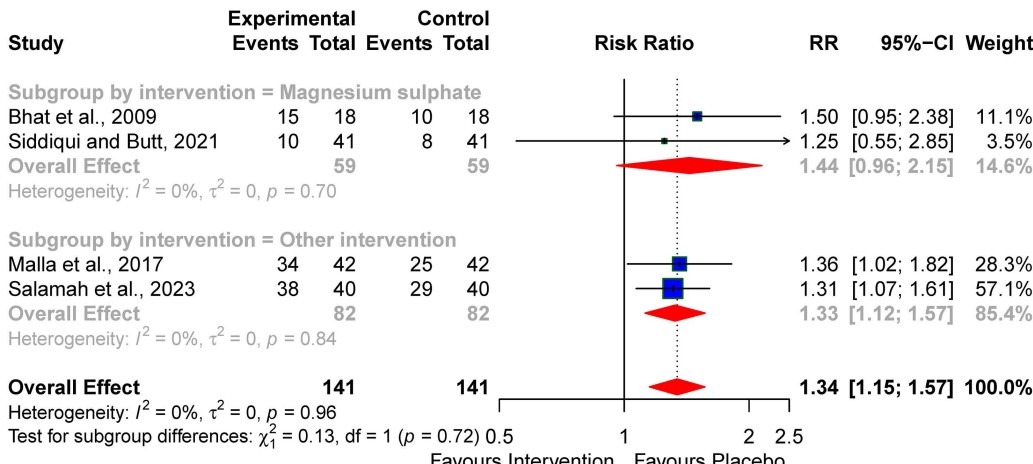

**Fig 7. Subgroup analysis of the effect of the pharmacological agent on the need for the anticonvulsant requirement at discharge.**

## Normal Neuroimaging features

The pooled effect from the seven studies that reported neuroimaging features after the pharmacologic interventions for treating perinatal asphyxia revealed significant normal neuroimaging features at discharge when used to treat perinatal asphyxia RR 1.55 (95% CI: 1.34–1.79, $I^2 = 0$%) (see Fig 8).

On subgroup analysis for magnesium sulphate the pooled RR was 1.49 (95% CI: 1.23–1.80, $I^2 = 0$%) normal neuroimaging features when used to treat perinatal asphyxia (Fig 9).

## Normal neurodevelopment outcomes

The studies that followed up the babies' neurologic outcomes at discharge or at one month demonstrated that the neuroprotective agents contributed to normal neurologic outcomes at discharge/one month of life, RR:1.78 (95% CI: 1.40–2.26; $I^2 = 0$%) (see Fig 10).

Subgroup analysis of the three studies that investigated the neurologic outcomes using magnesium sulphate at one month showed a pooled RR of 1.71 (95% CI: 1.30–2.25; $I^2 = 0$%) (Fig 11).

Our review shows better neurodevelopmental outcomes at 3 months with the use of topiramate (RR 2.23; 95% CI: 1.34–3.71), 19 months with the use of erythropoietin (RR 2.00; 95% CI: 1.28–3.13) (Fig 12)

## Publication bias and sensitivity analysis

According to Egger's test and funnel plots, publication bias was identified only for the outcome of successful initiation of oral feeding at discharge when the different types of interventions were pooled, but no publication bias was detected for the other outcomes. We used the trim-and-fill method to find that most of the examined outcomes remained consistent even after conducting sensitivity analyses. The funnel plots are shown in Supplementary file 6.

## Assessment of the certainty of a body of evidence using the GRADE framework

The GRADE approach [30] was used to assess each outcome's certainty across the domains outlined earlier. The findings are shown in Tables 5 and 6. The overall quality of evidence for MgSO4 was highly assessed for successful initiation of oral feeds by discharge and better neurologic outcomes at discharge/ one month of age. Only one study each employed melatonin and topiramate, hence the low sample sizes resulted in moderate evidence for their respective impacts on survival and better neurologic outcomes at 1 month.

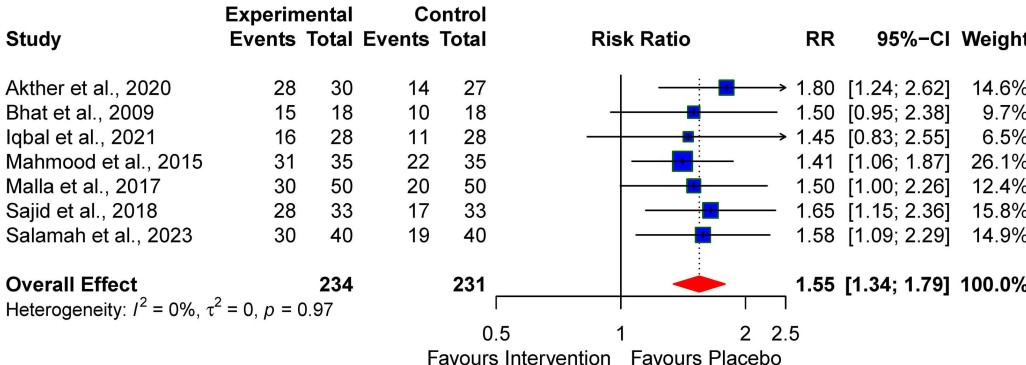

**Fig 8. Pooled effect of the neuroprotective agents on abnormal neuroimaging features at discharge/one month.**

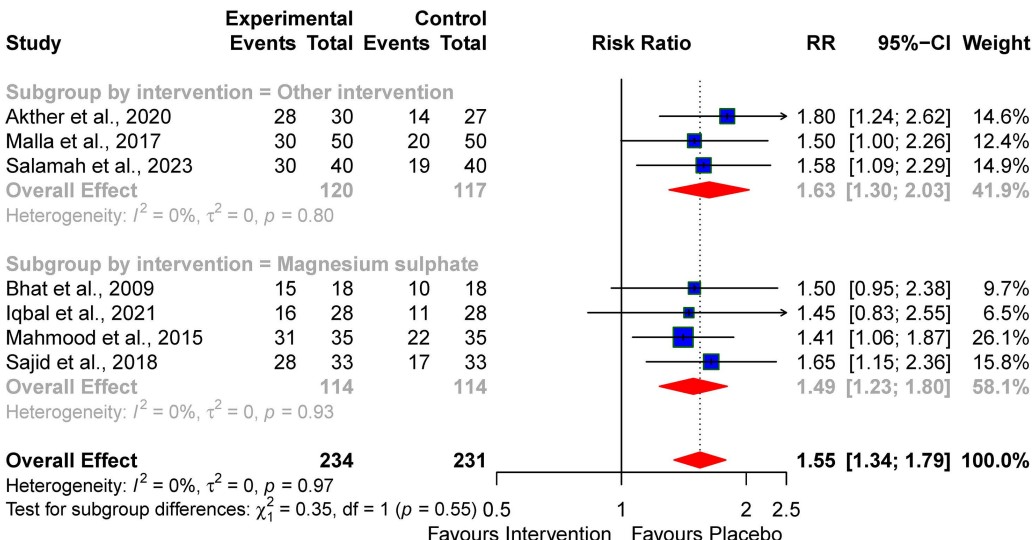

**Fig 9. Subgroup analysis of the effect of the neuroprotective agents on abnormal neuroimaging features at discharge/one month.**

## Discussion

This systematic review evaluated randomised control trials assessing the effectiveness of pharmacologic neuroprotective agents without the addition of therapeutic hypothermia for the treatment of PA in LILMIC. The findings suggest several pharmacologic agents show promising neuroprotective effects, improving survival and short- and long-term neurodevelopmental outcomes. However, the review also highlights a paucity of high-quality RCTs from LILMICs, on effective pharmacologic therapies for managing perinatal asphyxia in LILMIC, underscoring the urgent need for more research to identify viable and context-appropriate treatment options.

A notable observation was the temporal increase in research activity post-2014, reflecting a growing recognition of the limitations of therapeutic hypothermia in resource-limited settings [11,12]. This shift underscores the importance of alternative neuroprotective strategies where therapeutic hypothermia is either unavailable or ineffective [2].

Geographically, the studies included in this review were disproportionately conducted in Asia, particularly Pakistan, with no RCTs from sub-Saharan Africa—a region with some of the highest neonatal mortality rates due to perinatal asphyxia

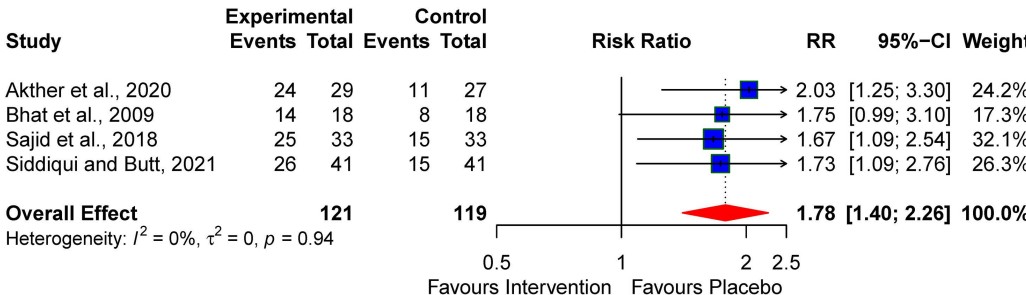

**Fig 10. Pooled effect of the neuroprotective agents on abnormal neurologic features at discharge/one month.**

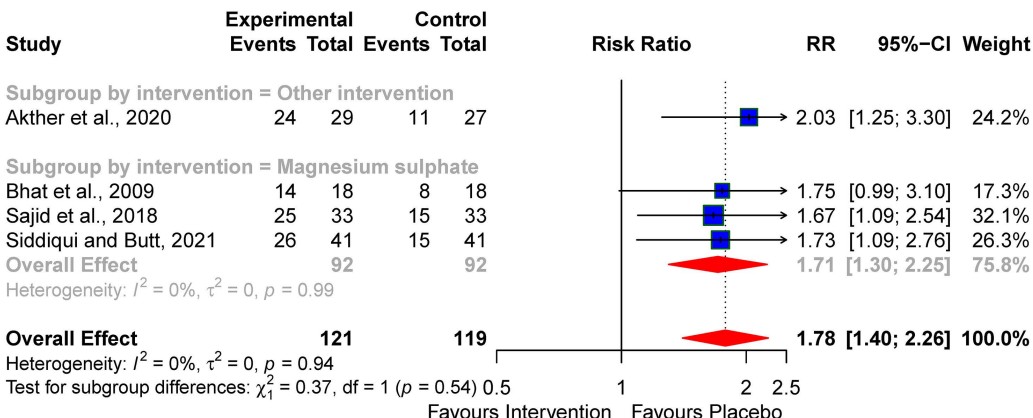

**Fig 11. Subgroup analysis of the effect of the neuroprotective agents on abnormal neurologic features at discharge/one month.**

[2] The sole study from Egypt limits the representativeness of the African context. This regional imbalance emphasises the need for more inclusive and diverse research to inform clinical practice across low-resource settings.

Magnesium sulphate was the most frequently studied among the neuroprotective agents evaluated—magnesium sulphate, melatonin, topiramate, erythropoietin, and citicoline. It consistently demonstrated improved neurological outcomes at various follow-up intervals, including 6 and 12 months [34,37]. Magnesium sulphate action is attributed to its mechanisms, such as calcium channel blockade [42–44] and N- methyl D-Aspartate (NMDA) receptor inhibition [37]—mechanisms that are hypothesised to confer neuroprotective benefits [18]. Magnesium sulphate was investigated in two-thirds of the included studies, and there is still heterogeneity in neuroprotective agents, as melatonin, topiramate, erythropoietin, and citicoline, have also been studied [18]. Each of these agents presents different mechanisms of action [18], offering a wide scope for future studies to explore their efficacy and safety in different contexts with a clinical diagnosis of perinatal asphyxia.

The meta-analysis identified several clinically relevant outcomes, including survival rates, initiation of oral feeding, absence of seizure activity at discharge, and neuroimaging findings. While magnesium sulphate showed consistent neuroprotective benefits in most outcomes, it did not significantly improve survival or neurodevelopment at 6 and 12 months. Conversely, melatonin was associated with improved survival—consistent with findings from high-income countries where it is often used as an adjunct to therapeutic hypothermia [45]. Erythropoietin, however, did not significantly reduce mortality, aligning with previous reviews by Pan *et al.* [46] and individual studies such as Malla *et al* [36].

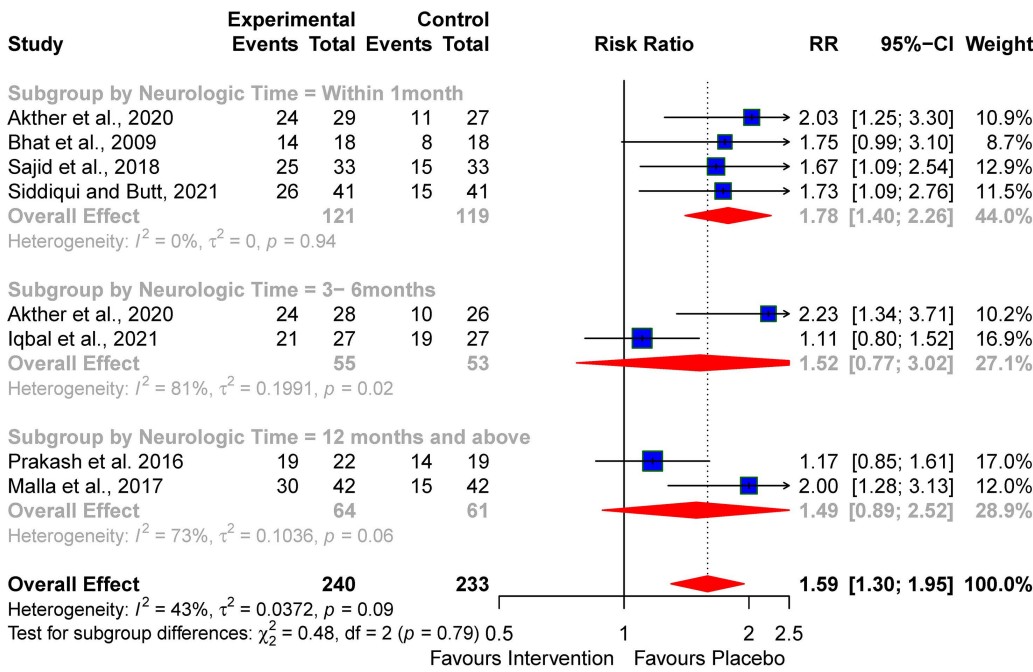

**Fig 12. Effects of the neuroprotective agents on abnormal neurodevelopmental features.**

Several included trials reported improved neurological outcomes, such as earlier initiation of oral feeding at discharge and reduced reliance on anticonvulsants [17,33,35,39,41]. However, substantial heterogeneity was observed in the pooled analysis, which was likely driven by the inconsistent effects of erythropoietin compared to the more favourable and consistent results observed with magnesium sulphate. This aligns with previous findings indicating stronger neuroprotective effects of magnesium [47]. The variability across studies may be attributed to differences in pharmacokinetics and mechanisms of action among the various agents.

In line with prior systematic reviews [48], most pharmacologic agents included in this review demonstrated improved seizure control—an essential clinical outcome in neonates with hypoxic-ischemic encephalopathy. In contrast, the meta-analysis by Pan *et al.* [46] reported that erythropoietin did not significantly reduce mortality or improve neurological outcomes. This discrepancy could be due to variations in dosing regimens or timing of administration across studies. Nonetheless, erythropoietin was not associated with an increased risk of adverse events, suggesting it is a relatively safe intervention. Its potential neuroprotective benefits may depend on specific factors such as dosage, timing of treatment, and the severity of the neonatal hypoxic insult.

Most of the pharmacologic agents reviewed were associated with improved seizure control, fewer abnormal neuroimaging findings at discharge or within the first month, and better neurological outcomes at one month of age. Longitudinal studies on individual agents—such as topiramate (up to 3 months), magnesium sulphate (6 and 12 months), and erythropoietin (up to 19 months)—reported sustained neurodevelopmental improvements. These findings suggest that pharmacologic interventions may confer long-term neuroprotective benefits beyond the acute phase of perinatal asphyxia, particularly in the cases of topiramate and erythropoietin. Such sustained effects are critical for improving long-term survival and quality of life—key endpoints in the treatment of perinatal asphyxia [2]. However, further research is needed to determine the optimal treatment protocols, including dosing, duration, and timing, to maximise the efficacy of these agents in clinical practice.

**Table 5. GRADE certainty assessment.**

| Certainty assessment | | | | | | | Summary of findings | | | | |
|---|---|---|---|---|---|---|---|---|---|---|---|
| Participants (studies) Follow-up | Risk of bias | Inconsistency | Indirectness | Imprecision | Publication bias | Overall certainty of evidence | Study event rates (%) | | Relative effect (95% CI) | Anticipated absolute effects | |
| | | | | | | | With Basic supportive care with or without placebo | With Pharmacologic neuroprotective agents | | Risk with Basic supportive care with or without placebo | Risk difference with Pharmacologic neuroprotective agents |
| **Survival with MgSO4** | | | | | | | | | | | |
| 271 (7 RCTs) | not serious | not serious | not serious | serious[a] | none | ⊕⊕⊕○ Moderate[a] | 102/134 (76.1%) | 114/137 (83.2%) | RR 1.08 (0.97 to 1.20) | 102/134 (76.1%) | 61 more per 1,000 (from 23 fewer to 152 more) |
| **Survival with melatonin** | | | | | | | | | | | |
| 80 (1 RCT) | not serious | not serious | not serious | serious[b] | none | ⊕⊕⊕○ Moderate[b] | 26/40 (65.0%) | 35/40 (87.5%) | RR 1.35 (1.04 to 1.74) | 26/40 (65.0%) | 228 more per 1,000 (from 26 more to 481 more) |
| **Successful initiation of oral feeding at discharge with MgSO4** | | | | | | | | | | | |
| 454 (5 RCTs) | not serious | not serious | not serious | not serious | none | ⊕⊕⊕⊕ High | 81/227 (35.7%) | 162/227 (71.4%) | RR 1.98 (1.63 to 2.40) | 81/227 (35.7%) | 350 more per 1,000 (from 225 more to 500 more) |
| **Absence of AED requirements at discharge with MgSO4** | | | | | | | | | | | |
| 118 (2 RCTs) | not serious | not serious | not serious | serious[a] | none | ⊕⊕⊕○ Moderate[a] | 18/59 (30.5%) | 25/59 (42.4%) | RR 1.44 (0.96 to 2.15) | 18/59 (30.5%) | 134 more per 1,000 (from 12 fewer to 351 more) |
| **Normal Neuroimaging features at Discharge with MgSO4** | | | | | | | | | | | |
| 228 (4 RCTs) | not serious | not serious | serious[c] | serious[a] | none | ⊕⊕○○ Low[a,c] | 60/114 (52.6%) | 90/114 (78.9%) | RR 1.49 (1.23 to 1.80) | 60/114 (52.6%) | 258 more per 1,000 (from 121 more to 421 more) |
| **Normal neurodevelopment outcomes at 1 month with Topiramate** | | | | | | | | | | | |
| 56 (1 RCT) | not serious | not serious | not serious | serious[b] | none | ⊕⊕⊕○ Moderate[b] | 11/27 (40.7%) | 24/29 (82.8%) | RR 2.03 (1.25 to 3.30) | 11/27 (40.7%) | 420 more per 1,000 (from 102 more to 937 more) |
| **Normal neurodevelopment outcomes at 1 month with MgSO4** | | | | | | | | | | | |
| 184 (3 RCTs) | not serious | not serious | not serious | not serious | none | ⊕⊕⊕⊕ High | 38/92 (41.3%) | 65/92 (70.7%) | RR 1.71 (1.30 to 2.21) | 38/92 (41.3%) | 293 more per 1,000 (from 124 more to 500 more) |

**CI:** confidence interval; **RR:** risk ratio; **MgSO4:** Magnesium sulphate

a. The width of the CI crosses unity in two of the articles

b. There was only one article, and the sample size is small.

c. One out of the four studies used cranial ultrasonography while the remaining three used cranial CT.

**Table 6. Summary of findings.**

| Outcomes | Anticipated absolute effects* (95% CI) | | Relative effect (95% CI) | № of participants (studies) | Certainty of the evidence (GRADE) | Comments |
|---|---|---|---|---|---|---|
| | Risk with Basic supportive care with or without placebo | Risk with Pharmacologic neuroprotective agents | | | | |
| Survival with MgSO4 | 761 per 1,000 | **822 per 1,000** (738–913) | **RR 1.08** (0.97 to 1.20) | 271 (7 RCTs) | ⊕⊕⊕○ Moderate[a] | MgSO4 probably improves survival in babies with perinatal asphyxia. |
| Survival with melatonin | 650 per 1,000 | **878 per 1,000** (676–1,000) | **RR 1.35** (1.04 to 1.74) | 80 (1 RCT) | ⊕⊕⊕○ Moderate[b] | Melatonin likely results in large increase in survival with melatonin, but there is a need for further RCTs to confirm this |
| Successful initiation of oral feeding at discharge with MgSO4 | 357 per 1,000 | **707 per 1,000** (582–856) | **RR 1.98** (1.63 to 2.40) | 454 (5 RCTs) | ⊕⊕⊕⊕ High | Use of MgSO4 results in large increase in successful initiation of oral feeding at discharge. |
| Absence of AED requirements at discharge with MgSO4 | 305 per 1,000 | **439 per 1,000** (293–656) | **RR 1.44** (0.96 to 2.15) | 118 (2 RCTs) | ⊕⊕⊕○ Moderate[a] | Use of MgSO4 likely results in an increase in absence of AED requirements at discharge. |
| Normal Neuroimaging features at Discharge with MgSO4 | 526 per 1,000 | **784 per 1,000** (647–947) | **RR 1.49** (1.23 to 1.80) | 228 (4 RCTs) | ⊕⊕○○ Low[a,c] | It is not certain if MgSO4 improves Neuroimaging features at discharge. There is a need for more studies with the use of a the same neuroimaging modality for further analysis. |
| Normal neurologic outcomes at 1 month with Topiramate | 407 per 1,000 | **827 per 1,000** (509 to 1,000) | **RR 2.03** (1.25 to 3.30) | 56 (1 RCT) | ⊕⊕⊕○ Moderate[b] | Topiramate probably improves neurologic outcomes at 1 month, but there is a need for further RCTs to confirm this. |
| Normal neurologic outcomes at 1 month with MgSO4 | 413 per 1,000 | **706 per 1,000** (537 to 913) | **RR 1.71** (1.30 to 2.21) | 184 (3 RCTs) | ⊕⊕⊕⊕ High | Use of MgSO4 results improves neurologic outcomes at 1 month of age. |

**Pharmacologic neuroprotective agents compared to basic supportive care with or without placebo for Perinatal asphyxia**

**Patient or population:** Perinatal asphyxia

**Setting:** LILMIC

**Intervention:** Pharmacologic neuroprotective agents

**Comparison:** Basic supportive care with or without placebo

*The risk in the intervention group (and its 95% confidence interval) is based on the assumed risk in the comparison group and the **relative effect** of the intervention (and its 95% CI).

**CI:** confidence interval; **RR:** risk ratio; **MgSO4:** magnesium sulphate

**GRADE Working Group grades of evidence**

**High certainty:** we are very confident that the true effect lies close to that of the estimate of the effect.

**Moderate certainty:** we are moderately confident in the effect estimate: the true effect is likely to be close to the estimate of the effect, but there is a possibility that it is substantially different.

**Low certainty:** our confidence in the effect estimate is limited: the true effect may be substantially different from the estimate of the effect.

**Very low certainty:** we have very little confidence in the effect estimate: the true effect is likely to be substantially different from the estimate of effect.

## Limitations of the review

The variability in sample sizes and outcome measures, the absence of multiple studies investigating the pharmacologic agents except for magnesium sulphate, and the regional imbalance of the included studies limit the generalizability of the findings for all low-income and lower-middle-income countries. Nevertheless, this review provides a solid basis for the promising use of pharmacological agents for the treatment of perinatal asphyxia in LILMICs.

## Conclusion

This review identifies magnesium sulphate as the most extensively studied pharmacologic neuroprotective agent, with emerging evidence supporting the potential of melatonin, erythropoietin, topiramate, and citicoline. By comparing these agents to standard supportive care, the review highlights their impact on critical outcomes such as survival and both short- and long-term neurodevelopment. These findings help fill an important knowledge gap and suggest scalable treatment options for resource-limited settings where therapeutic hypothermia is not feasible.

Importantly, this review lays the groundwork for future research and clinical protocol development—particularly in sub-Saharan Africa—by emphasizing the need to investigate long-term outcomes and evaluate the safety and efficacy of promising and novel neuroprotective agents for managing perinatal asphyxia.

## Supporting information

**S1 File. PRISMA 2020 checklist.**
(DOCX)

**S2 File. Search Strategy.**
(DOCX)

**S3 File. Data Extraction.**
(XLSX)

**S4 File. JBI Checklist.**
(XLSX)

**S5 File. Funnel plots for Outcomes used in the Meta-analysis.**
(DOCX)

**S6 File. Studies that might appear to meet the inclusion criteria, but which were excluded.**
(DOCX)

## Author contributions

**Conceptualization:** Victor Ayodeji Ayeni, Aisha Adam Abdullahi, Abdulmuminu Isah, Love Bukola Ayamolowo, Olunike Rebecca Abodunrin, Folahanmi Tomiwa Akinsolu.

**Data curation:** Victor Ayodeji Ayeni, Aisha Adam Abdullahi, Abdulmuminu Isah, Love Bukola Ayamolowo, Olunike Rebecca Abodunrin, Folahanmi Tomiwa Akinsolu.

**Formal analysis:** Victor Ayodeji Ayeni, Aisha Adam Abdullahi, Abdulmuminu Isah, Love Bukola Ayamolowo, Folahanmi Tomiwa Akinsolu.

**Funding acquisition:** Victor Ayodeji Ayeni, Aisha Adam Abdullahi, Abdulmuminu Isah, Love Bukola Ayamolowo, Timothy Adeyemi, Motunrayo Adebukunola Oladimeji, Ufuoma Shalom Ahwinahwi, Vitalis Ikenna Chukwuike, Oluchukwu Perpetual Okeke, Olajide Odunayo Sobande.

**Methodology:** Victor Ayodeji Ayeni, Aisha Adam Abdullahi, Abdulmuminu Isah, Love Bukola Ayamolowo, Olunike Rebecca Abodunrin, Folahanmi Tomiwa Akinsolu.

**Project administration:** Victor Ayodeji Ayeni, Oluchukwu Perpetual Okeke, Folahanmi Tomiwa Akinsolu, Olajide Odunayo Sobande.

**Resources:** Victor Ayodeji Ayeni, Aisha Adam Abdullahi, Abdulmuminu Isah, Love Bukola Ayamolowo, Timothy Adeyemi, Oluchukwu Perpetual Okeke, Olunike Rebecca Abodunrin, Folahanmi Tomiwa Akinsolu, Olajide Odunayo Sobande.

**Supervision:** Olunike Rebecca Abodunrin, Folahanmi Tomiwa Akinsolu, Olajide Odunayo Sobande.

**Validation:** Victor Ayodeji Ayeni, Aisha Adam Abdullahi, Abdulmuminu Isah, Timothy Adeyemi, Motunrayo Adebukunola Oladimeji, Ufuoma Shalom Ahwinahwi, Vitalis Ikenna Chukwuike, Olunike Rebecca Abodunrin, Folahanmi Tomiwa Akinsolu, Olajide Odunayo Sobande.

**Writing – original draft:** Victor Ayodeji Ayeni, Aisha Adam Abdullahi, Abdulmuminu Isah, Love Bukola Ayamolowo, Timothy Adeyemi, Motunrayo Adebukunola Oladimeji, Ufuoma Shalom Ahwinahwi, Vitalis Ikenna Chukwuike, Folahanmi Tomiwa Akinsolu.

**Writing – review & editing:** Victor Ayodeji Ayeni, Aisha Adam Abdullahi, Abdulmuminu Isah, Love Bukola Ayamolowo, Timothy Adeyemi, Motunrayo Adebukunola Oladimeji, Ufuoma Shalom Ahwinahwi, Vitalis Ikenna Chukwuike, Oluchukwu Perpetual Okeke, Olunike Rebecca Abodunrin, Folahanmi Tomiwa Akinsolu, Olajide Odunayo Sobande.

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
