## [Decision Letter · Decision Letter 0]

15 Oct 2025

Dear Dr. Ayeni,

Thank you for submitting your manuscript to PLOS ONE. After careful consideration, we feel that it has merit but does not fully meet PLOS ONE’s publication criteria as it currently stands. Therefore, we invite you to submit a revised version of the manuscript that addresses the points raised during the review process.

**ACADEMIC EDITOR: **

Statistical part should be clarified whether a staticians was a part of the the team.

We look forward to receiving your revised manuscript.

Kind regards,

Stefan Grosek, Ph.D., M.D.,

Academic Editor

PLOS ONE

Journal Requirements:

2 Thank you for stating the following financial disclosure:

“VAA, AAA, AI, LBA, TA, MAO, USA, VIC got a grant from Nigerian Institute of Medical Research Foundation for the work. Grant Number: NF-GMTP-24-152809”

Additional Editor Comments (if provided):

Dear Authors

Positive comments came out from both reviewers, however on of them wants to know if statistical approach is appropriate. Therefore I would like to ask you to explain whay this statistics was used and if statistician is part of of teams who prepared this paper.

Kind regards

Reviewers' comments:

Reviewer's Responses to Questions

**Comments to the Author**

1. Is the manuscript technically sound, and do the data support the conclusions?

Reviewer #1: Yes

Reviewer #2: Yes

2. Has the statistical analysis been performed appropriately and rigorously?

Reviewer #1: Yes

Reviewer #2: I Don't Know

3. Have the authors made all data underlying the findings in their manuscript fully available?

Reviewer #1: Yes

Reviewer #2: Yes

4. Is the manuscript presented in an intelligible fashion and written in standard English?

Reviewer #1: Yes

Reviewer #2: Yes

Reviewer #1: I congratulate the authors for a well and meticulously conducted research. I have the following points to make:

1. better keywords can be selected which are not a part of the title.

2. forest plot for different outcomes have not been provided.

3. Conclusion heading does not have complete text.

Reviewer #2: The paper by Victor Ayodeji Ayeni et al is a systematic review and metanalysis of eligible randomised controlled trials (RCTs) published between 2000 and 2024 on various pharmacologic neuroprotective agents for the treatment of perinatal asphyxia compared with placebo or standard care, excluding therapeutic hypothermia, in LILMICs. Data on survival and neurodevelopmental outcomes are analysed. The article fulfils some of the criteria for publication: the authors claim it is an original work and the results reported have not been published elsewhere. In the aspect of statistical analysis, I would very much appreciate if the article could be subjected to an expert on the field. The conclusions are presented in an understandable and appropriate fashion and are supported by the given data. The article is is written in good intelligible English. The research meets the applicable standards for the ethics of experimentation and research integrity. The article adheres to appropriate reporting guidelines and community standards for data availability.

Minor suggestions: why are there two different abbreviations used since I believe there could be only one: LMIC and LILMIC?

In the materials the Table 6 seems not in the appropriate form, it is not possible to read the text. Also, the references are missing, there is only minor part appearing at the end of the available materials.

**Do you want your identity to be public for this peer review?** For information about this choice, including consent withdrawal, please see our Privacy Policy

Reviewer #1: **Yes: ** Dr Ruchi Rai

Reviewer #2: No

---

## [Author Response · Author response to Decision Letter 1]

17 Oct 2025

1. Statistical part should be clarified whether a statistician was a part of the team.

Yes, one of the authors is indeed a Statistician and she was involved with the process of the systematic review from the beginning, through the meta-analysis and up till submission: Olunike Rebecca Abodunrin. The appropriate statistical methods were used and verified by the statistician on the team.

2. Why are there two different abbreviations used since I believe there could be only one: LMIC and LILMIC?

The review focused on low-income and lower middle-income countries (LILMICs) rather than low- and middle-income countries (LMIC). We decided to focus on LILMICs for better homogeneity of the included countries, as it was known that upper middle-income countries (which are part of LMIC but not part of LILMIC) have distinctive qualities, which may not be shared by pure LILMIC. However, LMICs were mentioned in the Background aspects of the Manuscript at points where we were referring to previous work and publications, which focused on LMIC (which included our subject area of LILMIC). The appearances of LMICs from referenced texts were found on pages 2-4.

3. In the materials the Table 6 seems not in the appropriate form, it is not possible to read the text. Also, the references are missing, there is only minor part appearing at the end of the available materials.

The font has been increased in size for both Tables 5 and 6, shown on pages 24-27

4. Line 33 on page 1, the spelling of “Biostatistics” was corrected.

5. Please state what role the funders took in the study.

6. Please include captions for your Supporting Information files at the end of your manuscript, and update any in-text citations to match accordingly.

The captions for the Supporting Information files are now included at the end of the manuscript, and supplementary files 1-6 (S1 – S6) have been re-submitted.

7. Reviewer 2: Has the statistical analysis been performed appropriately and rigorously?

Yes, one of the authors is indeed a Statistician and she was involved with the process of the systematic review from the beginning, through the meta-analysis and up till submission: Olunike Rebecca Abodunrin. The appropriate statistical methods were used and verified by the statistician on the team.

8. Reviewer 1: better keywords can be selected which are not a part of the title

The keywords section has now been updated as follows: perinatal asphyxia, neonatal encephalopathy, pharmacologic neuroprotective agents, magnesium sulphate, neurodevelopmental outcomes, low-income and lower-middle-income countries

9. Reviewer1: forest plot for different outcomes have not been provided.

These are shown in Figures 2 – 12.

10. Reviewer 1: Conclusion heading does not have complete text.

This has been checked.

11. I have confirmed the following: "The funders had no role in study design, data collection and analysis, decision to publish, or preparation of the manuscript."

12. I have included the following: "All data are in the manuscript and/or supporting information files".

12. The identifying information has been removed from the S3_Data extraction, and it has been resubmitted, along with S1, S2 and S4, because it was noted that the previous submission before review was retained because of similarity in names.

---

## [Decision Letter · Decision Letter 1]

14 Nov 2025

Pharmacologic neuroprotective agents for the treatment of perinatal asphyxia in low-income and lower-middle-income countries: a systematic review and meta-analysis of randomised controlled trials

PONE-D-25-16916R1

Dear Dr. Ayeni,

We’re pleased to inform you that your manuscript has been judged scientifically suitable for publication and will be formally accepted for publication once it meets all outstanding technical requirements.

Kind regards,

Stefan Grosek, Ph.D., M.D.,

Academic Editor

PLOS ONE

Additional Editor Comments (optional):

Dear Authors

Both reviewers found that all issuess have been addressed and implented in the paper. Therefore I suggest for acceptance of the paper.

Kind regards

Reviewers' comments:

Reviewer's Responses to Questions

**Comments to the Author**

Reviewer #1: All comments have been addressed

Reviewer #2: All comments have been addressed

2. Is the manuscript technically sound, and do the data support the conclusions?

Reviewer #1: Yes

Reviewer #2: Yes

3. Has the statistical analysis been performed appropriately and rigorously?

Reviewer #1: (No Response)

Reviewer #2: Yes

4. Have the authors made all data underlying the findings in their manuscript fully available?

Reviewer #1: Yes

Reviewer #2: Yes

5. Is the manuscript presented in an intelligible fashion and written in standard English?

Reviewer #1: Yes

Reviewer #2: Yes

Reviewer #1: (No Response)

Reviewer #2: (No Response)

**Do you want your identity to be public for this peer review?** For information about this choice, including consent withdrawal, please see our Privacy Policy

Reviewer #1: **Yes: ** Ruchi RaiNone

Reviewer #2: No

---

## [Editor Report · Acceptance letter]

PONE-D-25-16916R1

PLOS ONE

Dear Dr. Ayeni,

I'm pleased to inform you that your manuscript has been deemed suitable for publication in PLOS ONE. Congratulations! Your manuscript is now being handed over to our production team.

Kind regards,

on behalf of

Professor Stefan Grosek

Academic Editor

PLOS ONE